# Providing Diabetes Education through Phone Calls Assisted in the Better Control of Hyperglycemia and Improved the Knowledge of Patients on Diabetes Management

**DOI:** 10.3390/healthcare11040528

**Published:** 2023-02-10

**Authors:** Kanakavalli K. Kundury, Venugopal R. Bovilla, K. S. Prathap Kumar, Smitha M. Chandrashekarappa, SubbaRao V. Madhunapantula, Basavanagowdappa Hathur

**Affiliations:** 1Department of Health System Management Studies, JSS Academy of Higher Education & Research, Mysore 570015, Karnataka, India; 2Special Interest Group in Patient Care Management (SIG-PCM); JSS Medical College, JSS Academy of Higher Education & Research, Mysore 570015, Karnataka, India; 3Center of Excellence in Molecular Biology and Regenerative Medicine (CEMR, a DST-FIST Supported Center), Department of Biochemistry (a DST-FIST Supported Department), JSS Medical College, JSS Academy of Higher Education & Research, Mysore 570015, Karnataka, India; 4Department of Community Medicine, JSS Medical College, JSS Academy of Higher Education & Research, Mysore 570015, Karnataka, India; 5Special Interest Group in Cancer Biology and Cancer Stem Cells (SIG-CBCSC), JSS Medical College, JSS Academy of Higher Education & Research, Mysore 570015, Karnataka, India; 6Department of General Medicine, JSS Medical College & Hospital, JSS Academy of Higher Education & Research, Mysore 570015, Karnataka, India

**Keywords:** diabetes management, phone call-based education, HbA1C, randomized control trial, diabetes knowledge

## Abstract

**Purpose:** A recent single-arm pilot study from our group showed a significant decrease in HbA1C in Type-2 diabetes individuals provided with SMS and phone call-based education on glycemic control. Considering the preference of participants to phone call-based education, a randomized control trial (RCT) with parallel design was conducted to determine the impact of phone call-based diabetes educational intervention on the control of hyperglycemia and improvement in the knowledge about diabetes management. **Objectives:** To determine the impact of phone call-based educational intervention on the control of hyperglycemia and improvement in the knowledge about diabetes management. **Methodology:** The study was conducted for a period of 12 months on a total of 273 Type-2 diabetic patients (interventional group (n = 135); non-interventional group (n = 138)) who had provided consent to participate. Subjects in the case group received weekly phone calls on diabetes education; whereas the control group received no education. HbA1C investigations were carried out at baseline and at every fourth month until the completion of the study period for the subjects in both the groups. The impact of phone call-based education was measured by comparing HbA1C values as well as by measuring the questionnaire-based knowledge scores on diabetes management. **Results:** At the end of the study period, there was a significant reduction in HbA1C in 58.8% participants (n = 65) and a manifold (2–5-fold) increase in knowledge on diabetes management among participants in the case group (n = 110). However, no significant difference in HbA1C and knowledge score was observed in participants from the control group (n = 115). **Conclusion:** Phone call-based diabetes education is a viable option to empower patients for better management of Type-2 diabetes.

## 1. Introduction

Diabetes, a chronic metabolic disorder, is becoming an epidemic with increased incidence and prevalence globally [1]. In India, the number of individuals suffering from diabetes has exponentially increased from 26 million in the 1990’s to 66 million in 2016 and 77 million in the year 2020 [2,3]. Among 88 million diabetes cases that are reported in Southeast Asia, ~77 million cases (87.5%) are from India, making the country the hub of diabetics [4]. The International Diabetes Federation (IDF) reported that the prevalence of diabetes in India is about 8.9%, which is very close to the global prevalence of diabetes, i.e., 9.3% (in the year 2019) [4]. According to recent studies, this prevalence in India is expected to be much higher as many of the individuals are either unaware of this disease or not diagnosed [5,6]. The National Diabetes and Diabetic Retinopathy survey—2019 reported that the prevalence of diabetes in India remained at 11.8% in the previous four years (i.e., 2015–2019) [7]. In a separate report, the Longitudinal Ageing Study in India (LASI) observed more prevalence of diabetes among senior citizens and population living in urban areas [8]. About 11.5% of the individuals above 45 years of age were identified with high glycemic values [8]. In summary, the prevalence of diabetes is very high in India, and if immediate preventive steps are not taken, this number might increase further in the coming years.

Prior studies have shown that patient education on disease management practices and spreading will empower the affected individuals in the self-management of diabetes and associated disease conditions [9,10,11,12]. Due to the penetration of telecommunication tools and a strong possibility to extend their utility beyond the basic applications it is now feasible to provide health education more effectively even to individuals living in remote regions [13,14,15,16]. Very recently we conducted a pilot study by providing continuous diabetes education through Short Message Services (SMS) and phone calls (separately) and showed a significant reduction in HbA1C in ~47% of the study participants who had received periodic phone calls [17]. Since patients in the pilot study preferred phone calls, now in this Randomized Control Trial (RCT; with parallel design) we determined the effect of phone call-based educational intervention in mitigating the severity of diabetes (primary outcome—a decrease in HbA1C) as well as in improving the patients’ knowledge (increase in knowledge scores from baseline) about diabetes management.

## 2. Objectives

The objective of the study was to determine the impact of phone call-based educational intervention on the control of hyperglycemia and improving the knowledge about diabetes management practices in a RCT with parallel design.

## 3. Materials and Methods

### 3.1. Study Approval

The study was approved by Institutional Ethics Committee (IEC) of JSS Medical College, JSS Academy of Higher Education & Research (JSS AHER) (Protocol #: JSSMC/11/5976/2016-17) (Supplemental Information S1—IEC).

IEC of JSS AHER is a NABH recognized committee which operates under the guidelines of Indian Council of Medical Research (ICMR), Govt of India. The trial was retrospectively registered at International Standard Registered Clinical/Social Study Number (ISRCTN) registry, which is an approved clinical trial registry by WHO (Trial # ISRCTN67491047, accessed on 22 October 2022, https://doi.org/10.1186/ISRCTN67491047).

### 3.2. Study Site

The study was conducted for a period of 12 months from March 2019 until February 2020 at JSS Hospital, which is an 1800-bedded tertiary care teaching hospital located in the Mysuru district of Karnataka, India.

### 3.3. Research Site & Recruitment of Study Participants

The study was carried out at the general medicine department of the JSS Hospital. Participants were recruited into the study for a period of 1 month beginning from Jan 2019. The patients were tested for capillary blood glucose (CBG) in the Nutrition and Dietetics department and eligible participants were included into the study (n = 455). A total of 273 study subjects who met the inclusion criteria (listed below) were recruited in this study. Considering a pooled standard deviation of 1.1 with a power of 80% and 95% confidence interval (CI) the sample size calculated was 108 in the case and control groups. Considering the dropout rate of 20%, the final sample size in the control and case groups was determined to be 130. Participants who signed the informed consent form were randomly allocated (in a parallel design) to case (n = 135) and control groups (n = 138) by the principal investigator (Supplementary: Consort flow chart—Annexure #S1).

### 3.4. Criteria for Inclusion and Exclusion

The inclusion criteria of patients were (a) existing type 2 diabetes, (b) not suffering from any complex diseases such as cardiovascular diseases, terminal illness, and Cancer and Acquired Immunodeficiency Syndrome (AIDS); (c) non-pregnant; (d) ability to operate mobile phones and speak in the local language, i.e., Kannada or in English; (e) able to provide consent to participate in the study (Appendix A—Informed consent form English & Kannada). Participants who did not meet the abovementioned criteria were excluded from the study.

### 3.5. Study Method

Even though our pilot study results showed improvement in glycemic control with both SMS and phone call-based diabetes education, participants expressed their preference to receive phone calls rather than short messages [17]. Therefore, we selected a phone-call based mode of intervention in the randomized control trial (RCT) with parallel design. Participants in the case group were aware about the study materials, phone-call schedule, and orientation camps, but no such information was provided to participants in the control group.

At baseline, demographic information and knowledge about diabetes and diabetes management practices was captured (Appendix A: Demographic information & baseline data form; Appendix A: Schematic representation) from participants in the control and case groups. Next, a diabetologist conducted an orientation session to participants in the case group. The education material contained information about (a) diabetes and types of diabetes; (b) risk factors; (c) investigations to be carried out to confirm the diagnosis; (d) role of self-care in diabetes management that include diet control, medication adherence, periodic doctor visits, and investigations; and (e) monitoring the co-morbid conditions of diabetes (Appendix A: Educational material in Kannada & English). For better understanding, the orientation session was delivered in the local language. HbA1C was measured for all the participants (both in control and case groups) at baseline. A total of 4 camps were arranged to participants only in the case group, during the 12 month study period (baseline, fourth month, eighth month, and twelfth month, respectively). The education continued with weekly phone calls to the case group subjects (Appendix A: Phone call script in Kannada & English). Phone call logs were maintained in a separate sheet to ensure complete delivery of educational information (Appendix A: Weekly phone call log sheet). In addition, all individual participants’ diabetes management practice-related information was recorded (Appendix A: Data sheet for diabetes management practices).

To evaluate the impact of phone call-based education on the knowledge of participants in the case group, a total of 10 questions was asked at every fourth week (Appendix A: Knowledge assessment questionnaire). The questions were face validated before administering them in this study. Face validity is one of the content validity methods and refers to the degree to which the participants judge the questionnaire [18]. Although not a stronger way of validation, studies have shown that the face validity method motivates respondents to answer more truthfully [18]. Hence, in this study we used face validation to collect information pertaining to knowledge about diabetes. Each correct answer was awarded 1 point, while no point was given for an incorrect answer (Appendix A: Knowledge score card). The total scores of the individual participants were compared across time points to measure the fold increase in their knowledge from baseline. An average of 8 min was spent on each phone call per participant. Considering 4 calls per month, a total of 48 calls were made to each study participant in the case group for the study period of 12 months.

On the other hand, the subjects in the control group neither attended the orientation camps at every fourth month nor received any phone call-based education on diabetes care. HbA1C investigations were carried out for participants in the control group at baseline and at every fourth month throughout the study period. A total of 4 HbA1C investigations were carried out on each study subject in the case and control groups at baseline, the fourth month, eighth month, and twelfth month, respectively. The investigations were offered free of cost (as an incentive for participating in the study) to all the study subjects.

### 3.6. Blood Sample Collection and Analysis

About 3.0 mL of blood was collected from each study participant by a trained phlebotomist. HbA1C level was estimated using the Ion Exchange Chromatography (HPLC; D-10 Bio-Rad Hemoglobin Testing System) method.

### 3.7. Primary and Secondary Outcomes

Decrease in HbA1C value was the primary outcome of this study. Enhancement of knowledge about the diabetes and diabetes management practices were the secondary outcomes.

### 3.8. Statistical Analysis

The HbA1C results were subjected to One-Way ANOVA using Graph Pad Prism software to determine the impact of phone call-based education (for a period of 12 months) on the level of HbA1C and participants’ knowledge on diabetes management. A *p* value of <0.05 was considered significant. In addition, mean, standard deviation (SD), and standard error (SE) values of HbA1C at various time points were also calculated and compared. In order to determine whether the observed data were as per the expectations (no significant differences in different parameters between control and case groups at the baseline), a two-sided chi-square (Fisher’s exact test) analysis was performed.

## 4. Results

### 4.1. Demographic Details of Study Participants

About 83% of the type-2 diabetic population belongs to the age range of 41 to 70 years. Analysis of the study participants, at baseline, with diabetes revealed that 83.7% (113/135) individuals of “cases” and 82.6% (114/138) of “control” were in the age range of 41 to 70 years (Table 1; Appendix A, top panel). Further analysis showed that the HbA1C was more than 8.2 ± 1.86 in the control subjects and >9.7 ± 1.32 in the cases (Appendix A, top panel). A slightly higher HbA1C was observed in the population in the 60–70 age group compared with the 40–50 or 50–60 age groups. No significant differences in any of the parameters (age, age range, profession, duration of diabetes, and co-morbid conditions), were observed between control and case groups (*p* > 0.05; Two-sided Fisher’s Exact Chi-Square Test). The “*p*” value (0.046) between the control and case groups for sex showed a marginal significance, which could be due to the predominance of males in control group compared with the case group and, similarly, a predominance of females in the case group compared with control (Figure 1a; Table 1; and Supporting Excel Sheet #S1 containing statistical test results).

In the control group, the total number of study participants decreased from 138 at baseline to 115 over the 12 month period; a major decrease was observed in the age group 61–70 (36 at baseline to 27 at 12 months) (Appendix A). Similarly, in the case group, the total number of participants decreased from 135 at baseline to 110 at 12 months. A visible decrease was observed in the 61–70 age group (30 at baseline to 24 at 12 months) (Appendix A). The decrease in the number of participants was much higher in the 71–80 age range of both control (12 at baseline to 2 at 12 months) and case (10 at baseline to 2 at 12 months) groups. Potential reasons for this decrease in the number of participants in the older age groups could be (a) lack of interest in the study (controls 10%; cases 7.4%); (b) death (no deaths in controls; two deaths in cases); (c) unable to attend the camps (5% of cases); (d) poor health condition (controls 1.4%; cases 4.4%). No major variations were recorded in the HbA1C value of male and female participants in the control group throughout the study, indicating that these individuals have poor control over their HbA1C (Figure 1a,b). However, the HbA1C of male and female participants decreased from 9.98 and 10.2 at baseline to 7.76 and 7.86 at 12 months of intervention through mobile phone calls, respectively, indicating a better control over blood glucose (Figure 1a,b and Appendix A).

### 4.2. Analysis of the Impact of Phone Call-Based Education on Male and Female Participants

To determine whether the phone call-based education has similar or different effects in the male and female population, the collected data were analyzed, and the impact on HbA1C due to intervention was estimated. Analysis of the results showed that 44.4% (60 out of 135) and 55.6% (75 out of 135) were male and female, respectively, in the case group (Figure 2a). Among 60 male participants 55 had an HbA1C above 7.1. Similarly, among 75 female participants, 68 had an HbA1C above 7.1 (Figure 2a). In the control group 56.5% (78 out of 138) were male participants and 43.5% (60 out of 138) were female participants. Among 78 male participants 62 of them had an HbA1C above 7.1. Similarly, among 60 female participants 47 had an HbA1C above 7.1 (Figure 2a).

Sex-wise analysis of the data showed that the number of male and female participants decreased from baseline to 12 months. For instance, the number of male participants was 78 in the control group at baseline, which decreased to 64 at 12 months (Figure 2a,b, Table 2). Similarly, a decrease in female population was also observed (from 60 at baseline to 51 at 12 months) (Figure 2a,b, Table 2). Even in the case group, the number of male participants was decreased from 60 to 52 and the female participants number decreased from 75 at baseline to 58 at 12 months (Figure 2a,b, Table 2). Further analysis of the data showed that 33 out of 78 male subjects (42.3%) were in the HbA1C range of 7.1–9.0 at baseline. The percentage of males in this HbA1C range went up to 50.0% (32 out of 64) at the end of the study, indicating a slight improvement in the HbA1C control (Figure 2b and Table 2). The maximum number of female participants was observed in the HbA1C range between 7.1 and 9.0 at baseline of the control group (35%; 21/30). This number went up to 58.8% (n = 30/51) after 12 months of study (Figure 2b and Table 2).

Of the male participants in the case group at baseline, 38.33% (n = 23/60) were in the HbA1C range between 9.1 and 11.0, which decreased to 1.9% (n = 1/52) after 12 months of phone call-based intervention (Figure 2b and Table 2). Interestingly, the number of male participants in the HbA1C range between 5.1 and 7.0 (n = 3) and 7.1 and 9.0 (n = 18) increased significantly from baseline to 12 months post intervention, i.e., 5.1–7.0 (n = 18) and 7.1–9.0 (n = 33) (Figure 2b and Table 2). The maximum number of female participants (36%; n = 27/75) was observed in the HbA1C range of 7.1–9.0 at baseline. However, after 12 months of intervention, the percentage female population in this HbA1C range increased to 58.62% (n = 34/58), indicating that the phone call-based education improved the HbA1C of study participants (Figure 2b). Similarly, a significant increase in the number of female participants was observed even in the 5.1–7.0 HbA1C category (n = 3 at baseline to n = 21 at 12 months). In summary, the phone call-based intervention benefited the study participants irrespective of their sex and, hence, may be considered for further studies.

The mean value of years of diabetes indicated that all the study participants had diabetes for at least 6 years. While the male participants had 6.4 ± 5.6 years of diabetes in the case group, the control group had an average value of 8.1 ± 9.06 (Table 1). Likewise, the female study participants had an average of 6.2 ± 4.9 years of diabetes in the case group and 7.4 ± 7.6 years in the control group (Table 1).

### 4.3. Disease Management Practices of Study Participants at Baseline

In order to assess baseline knowledge about various diabetes management practices, a face-validated questionnaire measuring the (a) frequency of doctor visits; (b) frequency and type of blood investigation, (c) medication type and regimen; (d) adherence to dietary restrictions and prescribed medicines; (e) time spent on physical activity; and (f) frequency of foot and eye inspection was prepared and data were captured (Table 2). Analysis of the data showed visible differences in the control and case groups (Table 2). For example, only 24.4% (33/135 individuals) subjects recruited in the case group mentioned that they had visited the doctor once in a month, which is significantly different from the participants in control group, wherein 69.5% (96/138 individuals) participants marked that they had visited the doctor once in a month (Table 2; “*p*” value < 0.0001 by Chi-Square test for trend). Likewise, there were noticeable differences in the frequency of doctor visits (once in 3 months, 6 months, more than 6 months) by participants (Table 2; “*p*” value < 0.0001 by Chi-Square test for trend).

Similar differences were also observed between the case and control groups in (a) frequency of blood investigations (once in a month, once in 3 months, once in 6 months, or once in more than 6 months; “*p*” value < 0.0001 by Chi-Square test for trend); (b) type of blood investigation (fasting blood sugar—FBS, post-prandial blood sugar—PPBS, random and HbA1C; “*p*” value < 0.0001 by Chi-Square test for trend); (c) medication regimen (single oral hypoglycemic agent—OHA; multiple OHA, insulin and insulin with OHA; “*p*” value > 0.05 by Chi-Square test for trend); (d) adherence to medication (always, sometimes, never; “*p*” value < 0.0001 by Chi-Square test for trend); (e) time spent on physical activity (<= 15 min, 15–30 min, >30 min, and more than 30 min; “*p*” value < 0.0001 by Chi-Square test for trend); and (f) frequency of foot and eye inspections (always, sometimes, and never; “*p*” value > 0.05 by Chi-Square test for trend). Analysis of the abovementioned parameters showed that (a) none of the study participants (control and case groups) were aware of the HbA1C test; however, the majority of participants were aware of fasting (57.7% in the case group and 68.1% in the control group) and PPBS (46.6% in the case group and 68.8% in the control group) tests; (b) more than 50% participants are adhered to the prescribed medicines (54.07% in the case group and 82.6% in the control group); and (c) 48.1% (65/135) and 49.2% (68/138) participants, respectively, of the case and control groups always followed the dietary restrictions suggested by the doctor (Table 2 and Appendix A containing statistical test results).

### 4.4. Use of Anti-Hyperglycemic Agents Varied among Study Participants

Diabetes is generally treated with a wide variety of oral anti-hyperglycemic agents [19]. The type, dose, and regimen of anti-hyperglycemic agents vary not only from individual to individual but also based on the severity of disease [20,21,22]. Therefore, it is important to know about the medication history of study participants before initiating a study. Interestingly, in this study, the majority of study participants in the case and control groups were prescribed with either single OHA (40.7% in the case group and 63.04% in the control group) or multiple OHA (37.7% in the case group and 18.1% in the control group) (Figure 3 and Table 2). Analysis of the data showed that the phone call-based education reduced the HbA1C in each category, i.e., OHA—mono (9.7 ± 1.88 at baseline to 7.2 ± 0.74 after 12 months), OHA—multiple (10.2 ± 1.81 at baseline to 7.4 ± 0.82 after 12 months), insulin (9.6 ± 1.98 at baseline to 7.3 ± 0.76 after 12 months), and insulin with OHA (10.95 ± 0.91 at baseline to 7.7 ± 0.41 after 12 months) of the case group. The number of participants in each of these categories did not change much (Figure 3 and Table 2). However, in the control group, no major changes in the HbA1C levels were noticed among different categories. One of the reasons for the lack of improvement in the HbA1C of the control subjects could be non-adherence to medications by the participants.

### 4.5. Phone Call-Based Education Improved HbA1C Irrespective of Their Academic Statuss

To test whether the academic status of study participants has any impact on their response to phone call-based education and its impact on HbA1C level, the information about study participants’ education was collected and categorized them into those who had completed (a) school education; (b) college education; (c) post-graduate courses; and (d) professional education. Very few individuals reported that they had no formal education (10 and 7, respectively, in the case and control groups at baseline). Providing phone call-based education showed that all these individuals, irrespective of their formal education, exhibited significant improvements in their HbA1C levels after 12 months of education (Figure 4). However, no such improvement was observed in HbA1C in the control group (Figure 4).

### 4.6. Phone Call-Based Education Enhanced Participants’ Knowledge on Self-Management of Diabetes

Baseline data from our study showed that the participants had basic knowledge about diabetes and diabetes management practices. Further enhancement of existing knowledge through regular phone calls is likely to improve their diabetes management practices and overall health. Hence, in this study the diabetes education was provided through quarterly camps along with weekly phone call-based diabetes self-management education (about dietary restrictions, medication adherence etc.,) only to the participants in the case group. In addition, the impact of phone call-based education on participants’ knowledge was assessed during the fourth-week phone call of every month using a questionnaire consisting of 10 questions (1 mark each) (Appendix A).

The participants in control group were not provided with phone call-based education.

Analysis of the improvement in the knowledge scores after 12 months of study showed that out of 115 study participants in the control group who had completed the study, 73 (63.5%) had a knowledge score between 0 and 1.0, and 41 (35.65%) had a score between 1.1 and 2.0. Only one participant (0.85%) had a knowledge score in the 2.1–3.0 range (Figure 5). However, participants in the case group who had received phone call-based education throughout the study showed a significant improvement in their knowledge about diabetes (Figure 5). Eighty-four participants out of one-hundred and ten individuals (76.4%) who had completed the study had a knowledge score of 2.1–5.0 (Figure 5). The remaining 26 participants (23.6%) had a knowledge score greater than 7.1 (Figure 5). Therefore, it is clearly evident that the phone call-based education significantly improved the participants’ knowledge and thereby assisted in managing their disease.

### 4.7. Weekly Phone Call-Based Education for 12-Months Reduced HbA1C Percentage in about 58.58% Subjects

Earlier, our pilot study (in a pre- and post-design format) results showed a significant decrease in HbA1C percentage in 47% participants [17]. Based on this encouraging data and to further test and demonstrate the utility of phone call-based intervention in reducing the complications of type-2 diabetes, we planned a case–control study (in an RCT with parallel design) and tested the impact of providing phone call-based education on HbA1C percentage in both the control and case groups (Figure 6a,b). The majority of study participants (84/135, 62.22%) in the case group were in the average HbA1C range between 9.1 and 13.0 at baseline (Figure 6b). The remaining 51 individuals (38%) were in the HbA1C range between 5.1 and 9.0%. After 12 months of weekly phone call-based education, the number of individuals in the 5.1–9.0 HbA1C% range increased to 106 (out of 110 participants who had completed the entire study duration; 96.36%), indicating the benefits of providing phone call-based education in reducing the HbA1C in type-2 diabetics (Figure 6b). Similarly, after 12 months of intervention the number of participants in the HbA1C range between 9.1 and 15.0 significantly decreased to 4 out of 110 participants (3.64%), indicating that the phone call-based intervention reduced the HbA1C percentage in about 58.58% participants (62.90–3.64 = 58.58%).

Unlike participants in case group, the data in control group showed no observable difference between the HbA1C values at baseline and at the 12 month period, indicating that phone call-based intervention played an important role in the disease management (Figure 6a). The number of subjects in the control group at baseline was 83 out of 138 individuals (60.14%) who were in the HbA1C range of 5.1–9.0. The number of individuals in this HbA1C range remained the same even after 12 months (82 out of 115; 71.3%).

In summary, the phone call-based education appeared to be useful in reducing the HbA1C. Additional studies evaluating the similar intervention in a large community might further strengthen our findings. A systematic review and meta-analysis of five trials involving 953 patients reported that telephonic contact intervention was no more effective than standard clinical care in improving glycemic control [23]. However, authors of this meta-analysis concluded that telephonic contact intervention might be useful in low- and middle-income countries [23]. Therefore, study location and the participants’ economic status should also be considered while interpreting the data about the utility of telephonic contact intervention in improving knowledge about diabetes and bettering glycemic control.

## 5. Discussion

Type-2 diabetes (T2D) is one of the most important non-communicable diseases of great concern due to various reasons that include (a) associated co-morbidities such as increased cardiovascular diseases; (b) multi-organ failure, for example, renal failure; (c) susceptibility of effected individuals to various infections; and (d) early aging and death [19,24,25]. Even though there are effective T2D disease treatment options (such as medication) and management strategies available, controlling diabetes is challenging, as it is primarily a life-style disease and requires effective modifications in individuals’ lifestyle and dietary patterns [26,27,28]. Therefore, constant hand-holding and support is required to individuals suffering from diabetes. In this regard, many attempts have been made by several investigators to test the feasibility and evaluate the efficacy of mobile phone-based education to mitigate diabetes and diabetes-associated co-morbidities [29,30,31]. The current study aimed at identifying the impact of telephone call-based education, which resulted in decreased HbA1C values among 58.8% of the study participants, along with enhanced knowledge on diabetes self-management.

Mobile phone-based education is one of the best ways to communicate with individuals who are living in remote areas of the country [32,33]. According to recent statistics by Techarc (Technology Analytics Research and Consulting), a market research firm, India has more than 502 million smartphone users [34]. Using a mobile phone as a medium to communicate with individuals has many advantages such as (a) feasibility to communicate with individuals living even in very far and remote areas; (b) continuous monitoring and support; (c) individual-based interaction by live video/audio chatting; (d) no requirement for traveling and attending far-away places; (e) immediate solutions for some of the complications, etc. [35,36]. Moreover, mobile phone-based patient education, disease diagnosis, and e-treatment/m-treatment options are more preferred, especially in the case of older and physically challenged individuals [37,38].

In case of diabetes, educational intervention through mobile phones is one of the options tested for improving the quality of life and reducing the HbA1C in type-2 diabetics [39]. Mobile-based intervention strategies are also used to improve the knowledge about diabetes management practices [40,41]. In a recent systematic review, Garabedian L.F. et al., 2015 observed that mobile and smartphone (mHealth) technologies reduced HbA1C [42]. However, the authors of this study pointed out the need for additional investigations with longer follow-up to measure the efficacy of mHealth applications in mitigating diabetes and improving diabetes self-management practices [42].

Likewise, a study by Jeffrey, B. et al., 2019 compared T2D self-management practices between app users (n = 16) and non-app users (n = 14) and showed an improvement in T2D self-management practices and overall health [29]. However, critical barriers to the use of apps such as lack of knowledge and awareness about apps as health-care tools, internet connectivity issues, especially in rural settings, and perceptions of disease severity, etc., were reported by this study [29,43]. Furthermore, the app users, when compared with non-app users, exhibited a willingness to receive weekly text messages for their self-management but not just medication reminders [29]. This study recommends the inclusion of (a) educational features such as diabetes complications, hypoglycemic effects, etc.; (b) monitoring and tracking features to measure blood glucose, the impact on co-morbid conditions, etc.; and (c) nutritional features, such as the role of carbohydrates, etc. in the future apps. Considering some of these aspects, we executed a pilot study using short-messaging system (SMS) and phone call-based education for better management of T2D. The pilot study results showed an improvement in HbA1C in ~47% of study participants and patients preferred to have phone call-based education rather than receiving SMS [17]. Based on this data, and also by using the participants’ choice for phone calls over SMS, the current study was planned to further assess the efficacy of phone call-based education in the management of diabetes.

Russell E. Glasgow et al. reported a significant improvement in behavioral, psychological, and few biological outcomes when an internet-based self-management diabetes program (D-Net) was implemented for a period of 10 months in a population of 320 adult patients suffering from type-2 diabetes [44]. Rosal M.C., et al. conducted a community-based diabetes self-management educational intervention by providing information on diabetes knowledge, attitudes, and self-management skills through culturally specific and literacy-sensitive strategies for a duration of 6 months. The study demonstrated a significant decrease in HbA1C by 0.8 units in the intervention group at the 3 month and 6 month periods [45]. Further, an increase in physical activity and self-monitoring of glucose was reported among study participants (interventional group n = 15) compared with the non-interventional group (n = 10).

In a separate study, Maslakpak M.H. et al., 2017 reported that face-to-face and phone call-based interventions helped in bettering the patients’ fasting blood glucose, HbA1C, and lipid profile [46]. The authors of this paper tested the impact of face-to-face (n = 30) and phone call-based (n = 30) educational intervention for a period of 3-months, and compared the data (self-care, fasting blood sugar, HbA1C, cholesterol, and triglyceride) with a control (n = 30) group [46]. The results of this study showed improvement in the self-care (physical activity, glucose monitoring, foot care), adherence to dietary restrictions, and clinical outcomes (FBS, HbA1C, Lipid profile etc.) in face-to-face and phone-call based educational intervention groups compared with no intervention controls [46]. In summary, this study suggests the incorporation of family and patient education using low-cost phone calls as a part of standard patient care in the diabetes treatment protocols. Another study by Walker E.A., et al., 2011 compared the effectiveness of telephonic and print intervention over 1 year in improving HbA1C status, medication adherence, and self-care in 526 study participants whose HbA1C was >7.5% [47]. The authors of this study showed that 1 year of telephonic education implemented by health educators bettered HbA1C compared with print intervention [47]. An improvement in medication adherence was also reported in the telephonic education group, indicating the potency of this mode of intervention in reducing the complications of diabetes.

Sarayani, A., et al., 2018 evaluated the efficacy of a telephone-based intervention in a randomized control trial consisting of 100 type-2 diabetics [48]. The study comprised 16 telephone calls (by a pharmacist) for the intervention group and standard care for the control group. HbA1C was measured at baseline, 3 months after intervention, and at 9 months (follow-up). The results of this study showed a significant improvement in HbA1C in both the telephonic intervention group (from 8.0 at baseline to 6.97 at 3 months) and in the standard care control group (from 8.0 at baseline to 7.09 at 3 months) [48]. A pilot study involving nurse follow-up (12 weeks) with type-2 diabetes patients using mobile phone calls in promoting diabetes management yielded decreased HbA1C values among the case group in comparison with the control group participants [49]. The study concluded that patient follow-up through phone calls helps in better adherence to self-management of diabetes practices yielding short- and medium-term glycemic management by patients.

Telephone call-based follow-up education was found to improve patient adherence to diabetes care and also manage chronic disease conditions as per the study results of Brown-Deacon et al. [50]. According to them, along with the standard care, handholding the patients with continuous education through phone calls showed a clinically significant improvement in HbA1C values when compared with the other group of participants who received standard diabetes care alone. Demonstrating the effectiveness of telephone call-based education in better management of diabetes, Gallegos-Cabriales et al. tested automated phone call services as a self-care support method for adult Mexican patients with type-2 diabetes. The results revealed a decrease in HbA1C values when compared with baseline among those who received phone calls along with reduced depression scores when compared with the ones who did not receive the follow-up calls [51].

Many similar studies have also demonstrated the utility of telephone-based educational interventions in mitigating the HbA1C and improving patient medication adherence, following dietary restrictions, and self-testing. Similar observations were also made in the current study. In summary, the phone call-based educational intervention decreased HbA1C in 58.8% of participants with 2- to 5-fold increases in their knowledge, whereas no significant changes were observed among control group subjects. In conclusion, intervention through telephone-based education is likely to improve the health of individuals with diabetes. Hence, future practices can also consider including telephone-based education as a support to the standard-care protocol.

## 6. Study Limitations

Even though the study was executed with the recommended number of participants (as per statistical considerations), it would have been much stronger if a greater number of participants from different regions would have participated in this study. It would have been more appropriate if we had considered measuring lipid profile and other co-morbid conditions along with HbA1C, as diabetes is known to cause many other complications in individuals.

## 7. Conclusions

In conclusion, the results of the current study showed that (a) continuous education on diabetes management can impact patient knowledge and encouraged the study participants to adopt diabetes self-management practices; (b) diabetes education through phone call was the preferred modality by study participants compared with SMS; (c) phone call- and SMS-based diabetes educational intervention achieved a significant reduction of HbA1C in ~47% of study participants; and (d) the randomized controlled trial showed a decrease in HbA1C in ~58% of the study participants who had received phone-call based education (case group). In conclusion, phone call-based education is a viable option for improving the outcomes of diabetes treatment regimens.

## Figures and Tables

**Figure 1 healthcare-11-00528-f001:**
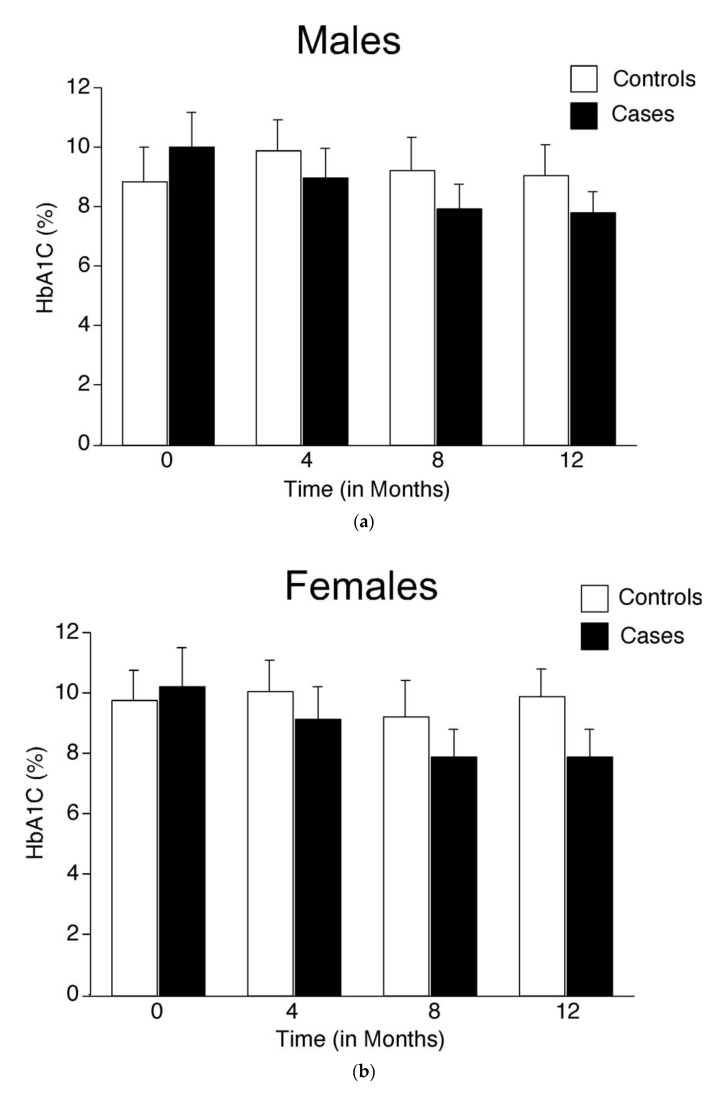
(**a**): Variations in HbA1C (%) values among male participants of the study across different time points: Analysis of the impact of providing phone call based education to male participants showed a visible decrease in HbA1C value at 12 months compared with the control group as well as the baseline HbA1C value within the group. The HbA1C value between the control and case groups was not significant at all time points (*p* > 0.05 by Student’s *t*-Test). However, a non-significant decrease in the HbA1C was observed in the case group at the fourth, eighth, and twelfth months of intervention (*p* > 0.05 by Student’s *t*-Test); (**b**): Variations in HbA1C (%) values among female participants of the study across different time points: Analysis of the impact of providing phone call-based education to female participants showed a visible decrease in HbA1C value at the eighth and twelfth months compared with the control group. The HbA1C value between the control and case groups was not significant at all time points (*p* > 0.05 by Student’s *t*-Test). However, a non-significant decrease in the HbA1C was observed in the case group at the fourth, eighth, and twelfth months of intervention (*p* > 0.05 by Student’s *t*-Test).

**Figure 2 healthcare-11-00528-f002:**
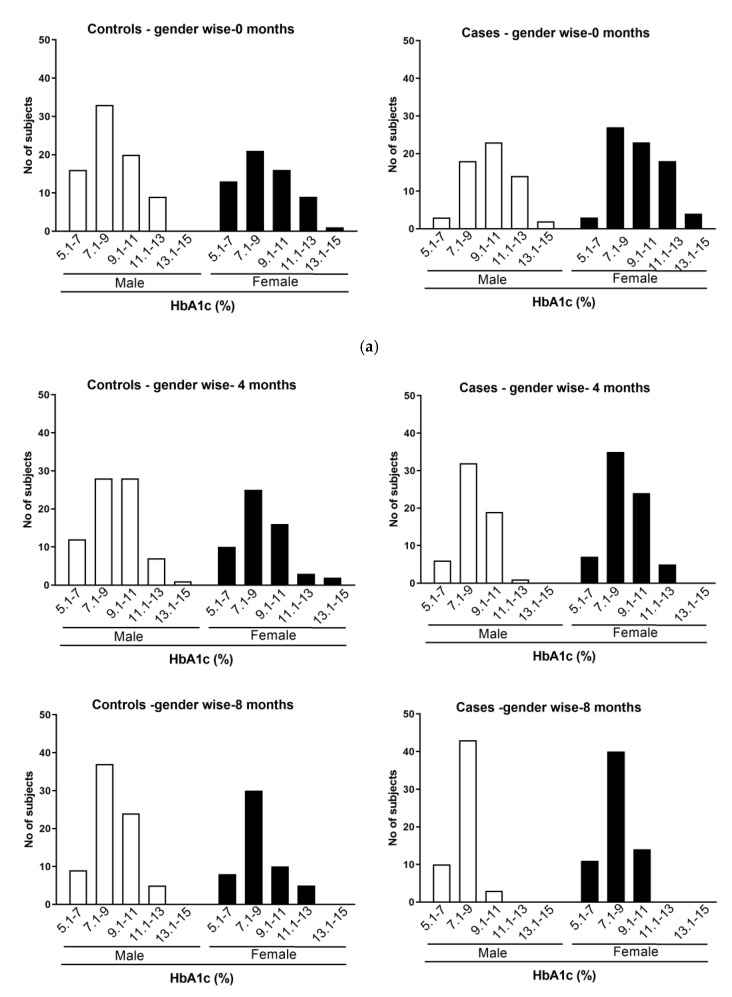
(**a**): Sex-wise distribution of study participants in the control and case groups in RCT: At baseline, the control and case groups of RCT comprised 138 and 135 study participants, respectively. Whereas males (56.5%; n = 78/138) were more predominant in the control group, the females (55.5%; n = 75/135) were more in the case group. (**b**): Distribution of male and female study participants in the control and case groups in the RCT. The male and female study participants of the control and case groups were distributed into different categories based on their HbA1C values. There was a significant difference in the proportion of study participants compared to base line at various time points (4, 8 & 12) at various categories of HbA1c levels among males and females (*p* < 0.05 by Chi Square test).

**Figure 3 healthcare-11-00528-f003:**
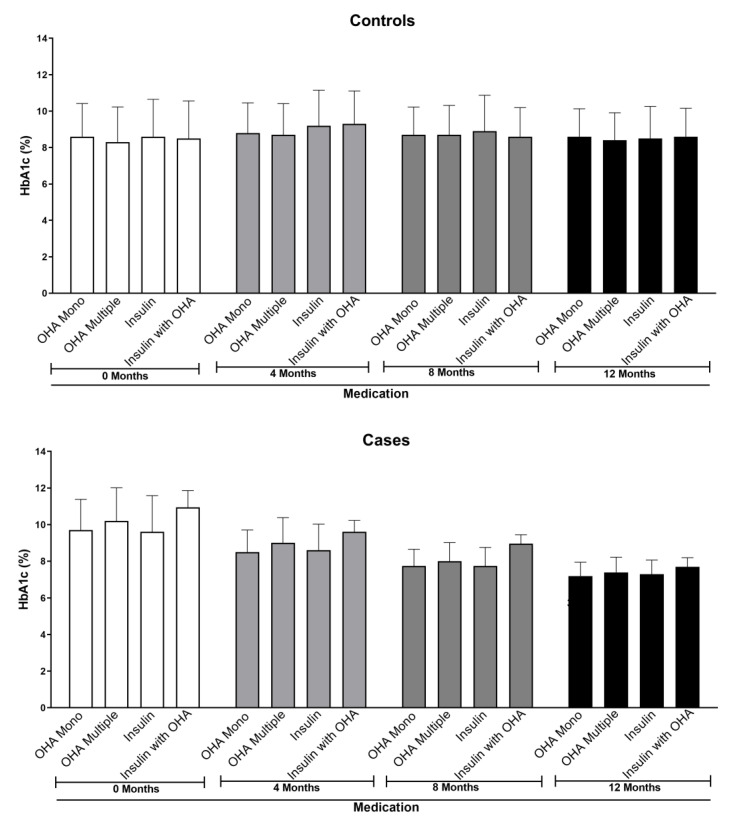
Use of anti-hyperglycemic agents and intervention through phone call-based education on HbA1C variations among the study participants. Phone call-based education assisted in reducing the HbA1C in diabetics as evidenced by a visible decrease in the HbA1C only in participants in the case group compared with the ones in the control group. The observed differences in the case group compared with baseline HbA1C were significant, especially the OHA, OHA—multiple, and insulin categories at 4, 8 & 12 Months (*p* < 0.05; One-Way ANOVA, Tukey’s multiple comparison test). However, the insulin and OHA group showed a significant difference only at the 12 month period.

**Figure 4 healthcare-11-00528-f004:**
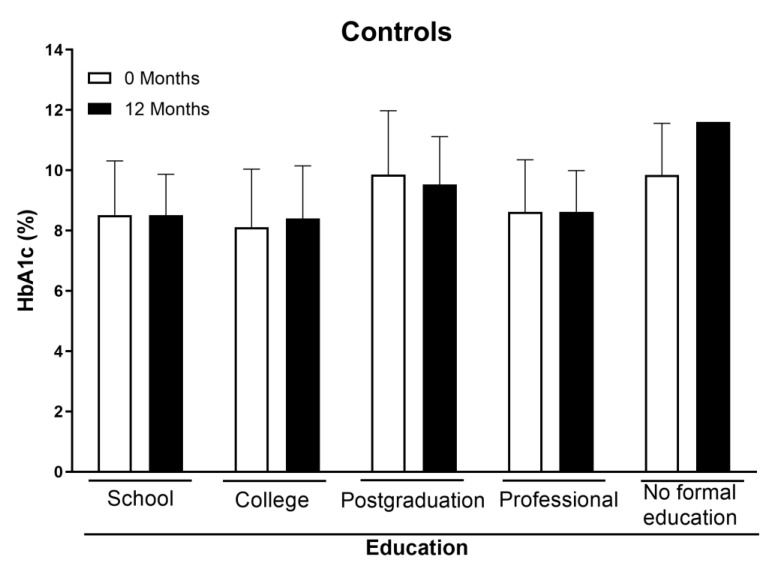
Phone call-based education decreased HbA1C level irrespective of the study participants’ education status. In order to test whether the phone call-based education-mediated beneficial effect varied with participants’ education status, the impact on HbA1C was compared with participants’ education status both in the control and case groups. Analysis of the data showed that providing phone call-based education decreased the HbA1C levels irrespective of participants’ education status, indicating that participants’ background education had no impact. It also demonstrates that the provided phone call script uniformly helped the individuals in bettering their diabetes status. Analysis of the data using statistical tools showed that the differences in HbA1C value between the 0 (i.e., baseline) and 12th month data was significant in the case group who had school-level, college-level education and professional education (*p* < 0.05, *t* test). No significant differences were observed in the control group.

**Figure 5 healthcare-11-00528-f005:**
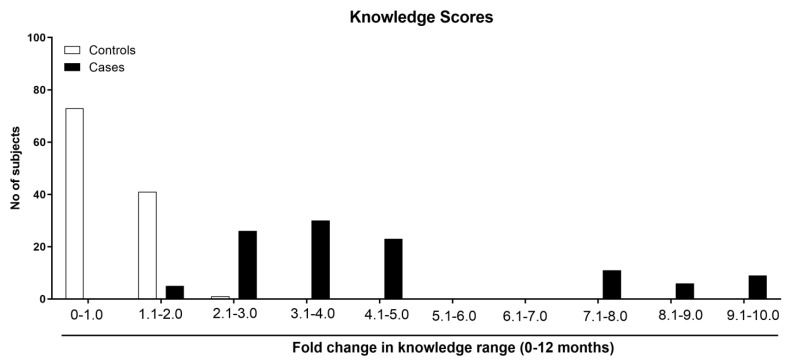
Phone call-based education improved participants’ knowledge about diabetes. In addition to reducing the HbA1C, providing phone call-based education enhanced participants’ knowledge, as evidenced by the presence of a greater number of study participants in the higher knowledge score groups. Participants in the control group who had not received any phone call-based education remained in the lower knowledge score category, indicating that the phone call-based education improved the knowledge of the participants and thereby helped them to mitigate the complications of diabetes.

**Figure 6 healthcare-11-00528-f006:**
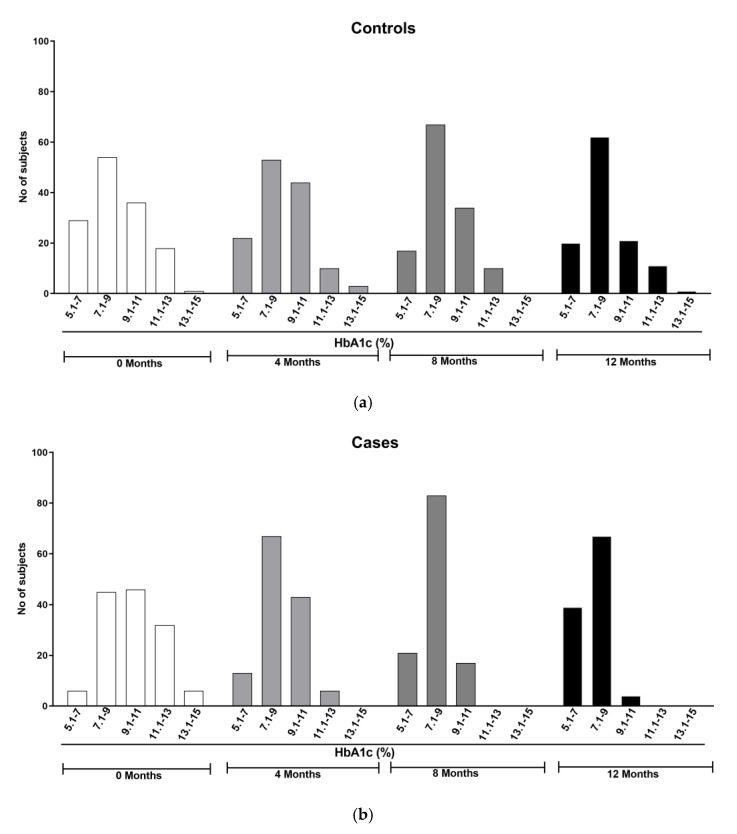
(**a**,**b**): Comparison of HbA1C values among the control and case groups across the study period. Providing education through phone calls increased the number of individuals in the HbA1C range from 7.0 to 13.0 at baseline to 5.0 to 9.0 after 12 months of intervention. No such increase in the number of participants in the lower HbA1C range was observed in the control group, who had not received phone call-based education. The obtained data were significant especially in the range of HbA1C 7.0-9.0 in the case group (*p* < 0.05; Tukey’s multiple comparison test).

**Table 1 healthcare-11-00528-t001:** Demographic information of the study participants in the parallel design RCT (please refer to Supporting Excel Sheet #S1 containing the statistical test results).

Demographics	Case Group				Control Group			
**Sex**	**N = 135**		**% of Total**		**N = 138**		**% of Total**	
Male	60		44.5		78		47.2	
Female	75		55.6		60		52.9	
**Age**	**Mean value**				**Mean value**			
Male	53.5 ± 11.1				56.1 ± 11.0			
Female	54.3 ± 9.0				56.9 ± 11.6			
**Age Range (Years)**	**N = 135**		**%**		**N = 138**		**% of total**	
	**M**	**F**	**M (N/60) × 100**	**F (N/75) × 100**	**M**	**F**	**M (N/78) × 100**	**F (N/60) × 100**
21–30	1	0	1.67	0	0	0	0	0
31–40	6	4	10	5.4	6	4	7.7	6.7
41–50	16	23	26.7	30.7	22	14	28.3	23.4
51–60	20	24	33.4	32	18	24	23.1	40
61–70	12	18	20	24	28	8	35.9	13.4
71–80	4	6	6.7	8	3	10	3.9	16.7
81–90	1	0	1.7	0	1	0	1.3	0
**Location**	**N = 135**		**% of total**		**N = 138**		**% of total**	
Mysuru	103		76.3		138		100	
Mandya	20		14.9		None		None	
Chamrajanagara	7		5.2		None		None	
Hassan	5		3.8		None		None	
Others	0		0		None		None	
**Domicile**	**N = 135**		**% of total**		**N = 138**		**% of total**	
Urban	94		69.7		138		100	
Semi-urban	18		13.4		None		None	
Rural	23		17.1		None		None	
**Profession**	**N = 135**		**% of total**		**N = 138**		**% of total**	
Agriculture	21		15.6		4		2.9	
Professional employment	23		17.1		37		26.9	
Service industry	29		21.5		31		22.5	
Homemaker	42		31.2		42		30.5	
Self-employment	11		8.2		17		12.4	
Retired	5		3.8		7		5.1	
Others	4		2.9		0		0	
**Duration of diabetes: Range in Years (Mean ± SDEV)**								
**Sex**	**Mean Value**				**Mean Value**			
Male	0.25–30 (6.4 ± 5.6)				0.50–41 (8.1 ± 9.06)			
Female	0.083–24 (6.2 ± 4.9)				0.083–35 (7.4 ± 7.6)			
**Diabetes associated co-morbid conditions**								
**Co-morbid conditions**	**N = 135**		**% of total**		**N = 138**		**% of total**	
Hypertension	24		17.8		26		18.8	
Retinopathy	45		33.4		46		33.4	
Renal disorders	05		3.8		02		1.5	
Foot ulcers	28		20.8		12		8.7	
Heart complications	12		8.9		16		11.6	
Neural disorders	03		2.3		05		3.7	
None of the above	56		41.5		67		48.6	

**Table 2 healthcare-11-00528-t002:** Participants’ disease management practices in RCT (please refer to Supporting Excel Sheet #S2 containing the statistical test results).

	Case Group (N = 135)		Control Group (N = 138)	
**Frequency of Doctor Visit**				
**Period**	**N = 135**	**% of Total**	**N = 138**	**% of Total**
Once in a month	33	24.4	96	69.5
Once in 3 months	99	73.3	26	18.8
Once in 6 months	03	2.2	10	7.2
Once in more than 6 months	0	0	06	4.3
**Frequency of blood investigation**				
**Period**	**N = 135**	**% of total**	**N = 138**	**% of total**
Once in a month	26	19.2	93	67.3
Once in 3 months	86	63.7	23	16.6
Once in 6 months	10	7.4	10	7.2
Once in more than 6 months	15	11.1	12	8.6
**Type of blood investigations**				
**Investigation**	**N = 135**	**% of total**	**N = 138**	**% of total**
Fasting	78	57.7	94	68.1
Post-prandial blood sugar (PPBS)	63	46.6	95	68.8
Random	17	12.5	19	13.7
HbA1C	0	0	14	10.1
**Medication regimen**				
**Medication**	**N = 135**	**% of total**	**N = 138**	**% of total**
Single OHP	55	40.7	87	63.04
Multiple OHP	51	37.7	25	18.1
Insulin	27	20	12	8.6
Insulin with OHP	02	1.4	14	10.1
**Adherence to dietary restrictions**				
**Adherence**	**N = 135**	**% of total**	**N = 138**	**% of total**
Always	65	48.1	68	49.2
Sometimes	51	37.7	52	37.6
Never	19	14.07	18	13.04
**Adherence to medication**				
**Adherence**	**N = 135**	**% of total**	**N = 138**	**% of total**
Always	73	54.07	114	82.60
Sometimes	42	31.1	14	10.14
Never	20	14.8	10	7.2
**Time spent on physical activity**				
**Time (minutes)**	**N = 135**	**% of total**	**N = 138**	**% of total**
<= 15 min	38	28.1	18	13.04
15–30	61	45.1	30	21.7
>30	17	12.5	68	49.2
No physical activity	19	14.07	22	15.9
**Frequency of foot inspection**				
**Frequency**	**N = 135**	**% of total**	**N = 138**	**% of total**
Always	28	20.7	56	40.5
Sometimes	68	50.3	26	18.8
Never	39	28.8	56	40.5
**Frequency of eye inspection**				
**Frequency**	**N = 135**	**% of total**	**N = 138**	**% of total**
Always	28	20.7	56	40.5
Sometimes	68	50.3	26	18.8
Never	39	28.8	56	40.5

## Data Availability

The data presented in this study are available on request from the corresponding author. The data are not publicly available due to privacy and ethical reasons.

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
