# Peer review of "Providing Diabetes Education through Phone Calls Assisted in the Better Control of Hyperglycemia and Improved the Knowledge of Patients on Diabetes Management"

_healthcare, 2023, doi:10.3390/healthcare11040528_

Round 1

Reviewer 1 Report

This study was a randomized controlled clinical trial assessing the efficacy of regular educational phone calls to improve self management of Type 2 diabetes. This clinical trial took place at a single hospital in India. Overall, this trial aimed to evaluate a new way of educating the public on diabetes management. However, this study also had some pitfalls including missing essential controls and statistical comparisons. 

#1 In the second paragraph of the Introduction, the statements regarding the pilot study are somewhat confusing and seem to imply this current manuscript describes that study instead of the new clinical trial. The authors should clarify the results discussed in this section are from a previous study.

#2 The authors state that participants in this study were aware of the materials and orientations camps but were not aware of what group they were assigned to. However, after not participating in an orientation camp and not receiving a phone call, the control group becomes aware they are in the control group and vise-versa. How did the authors prevent this or do you think this may have affected the results?

#3 As stated in the paper, the control group did not attend orientation, whereas the case group did. How can you be sure the results obtained were from the subsequent phone calls and not that initial orientation? 

#4 On line 144, the term "face validation" is used. What is this? No details or explanation was provided. 

#5 All the tables require statistical tests and resulting p-values comparing the values. In Table 1, p-values are required to ensure the groups were not significantly different, and in Table 2, p-values are required to show that the groups were significantly different. 

#6 In Table 1, the demographics are presented as total n and % of total. However, the % of total should be renamed to clarify that these percentages reflect the % of the total number of males and total number of females, not the total number of participants. 

#7 The graphs do not display the data in a clear and concise way. It is unclear that the numbers above the bar graphs indicate the HbA1C levels. Since HbA1C levels were the primary outcome, they should be displayed on the y-axis instead of number of subjects. The number of subjects can be included as "n=x" above each bar. In addition, there should not be more than one x-axis on any graph. 

#8 Throughout all the results, the changes observed were from baseline to the various timepoints. The authors did not include a comparison of these changes between the case and control groups. The authors should do statistical tests on these changes to show the significance between the two groups. 

#9 The authors state that slightly higher HbA1C levels were observed in 60-70 year old population. However, this age group also contained more males than females. Are there known sex differences that may contribute to the higher HbA1C levels?

#10 On line 253, the authors state that the HbA1C levels increased to 58.2% from baseline in the case group and that this indicated that the phone calls were successful. However, this conclusion cannot be made unless this increase was compared to the change in HbA1C levels in the control group from the baseline to the end of the study. In addition, all the changes seen in the case group from baseline should be compared to changes seen in the control group to accurately assess the phone-call education system. 

#11 In line 272, it states ...data showed a visible difference in the control and diabetic subjects." However, both the control and case groups included diabetic subjects. 

#12 In section 3.3, are the baseline characteristics significantly different between the two groups, as suggested? If so, how do these differences affect the results? P-values here would help clarify these differences. 

Author Response

Comments and Suggestions for Authors:  This study was a randomized controlled clinical trial assessing the efficacy of regular educational phone calls to improve self-management of Type 2 diabetes. This clinical trial took place at a single hospital in India. Overall, this trial aimed to evaluate a new way of educating the public on diabetes management. However, this study also had some pitfalls including missing essential controls and statistical comparisons. 

Query #1 In the second paragraph of the Introduction, the statements regarding the pilot study are somewhat confusing and seem to imply this current manuscript describes that study instead of the new clinical trial. The authors should clarify the results discussed in this section are from a previous study.

Response: Authors have clarified this by rewriting the second paragraph

Query #2: The authors state that participants in this study were aware of the materials and orientations camps but were not aware of what group they were assigned to. However, after not participating in an orientation camp and not receiving a phone call, the control group becomes aware they are in the control group and vise-versa. How did the authors prevent this or do you think this may have affected the results?

Response:  Authors thank the reviewer for this comment.  Participants in the control group are not aware about the phone call-based education and study materials.  This information is updated in the text of the revised manuscript

Query #3: As stated in the paper, the control group did not attend orientation, whereas the case group did. How can you be sure the results obtained were from the subsequent phone calls and not that initial orientation? 

Response:  As mentioned in the introduction this study was designed based on our pilot study outcomes demonstrating the combination of orientation and phone calls/SMS that resulted in the improvement in the diabetes management and HbA1C.  Accordingly, in this RCT, one orientation camp was conducted in the beginning of this study to participants in the case group.  Orientation alone had very minimal or no significant effect on patients’ knowledge about diabetes or on patients’ HbA1C level.   

Query #4: On line 144, the term "face validation" is used. What is this? No details or explanation was provided. 

Response:  Face validation is one of the methods to validate a questionnaire.  Details about face validation are provided in the revised manuscript

Query #5: All the tables require statistical tests and resulting p-values comparing the values. In Table 1, p-values are required to ensure the groups were not significantly different, and in Table 2, p-values are required to show that the groups were significantly different. 

Response:  As per reviewer suggestion, the p-values have been included as supporting Excel sheets in the revised submission

Query #6: In Table 1, the demographics are presented as total n and % of total. However, the % of total should be renamed to clarify that these percentages reflect the % of the total number of males and total number of females, not the total number of participants. 

Response: Authros thank the reviewer for this suggestion.  The recommended changes were included in the revised submission. 

Query #7: The graphs do not display the data in a clear and concise way. It is unclear that the numbers above the bar graphs indicate the HbA1C levels. Since HbA1C levels were the primary outcome, they should be displayed on the y-axis instead of number of subjects. The number of subjects can be included as "n=x" above each bar. In addition, there should not be more than one x-axis on any graph. 

Response:  The graphs have been restructured as per the suggestion of the reviewer

Query #8: Throughout all the results, the changes observed were from baseline to the various timepoints. The authors did not include a comparison of these changes between the case and control groups. The authors should do statistical tests on these changes to show the significance between the two groups. 

Response: Even though it is good to compare the data between control and case groups at every timepoint it will be difficult to show the data for all parameters in the manuscript.  Such comparisons were shown in the Ph.D. thesis of the first author Dr. Kanakavalli.  Since the objective of this study is to determine the changes in HbA1C and knowledge scores before and after phone call based education and orientation, in the main manuscript we have shown only the data pertaining to the differences between baseline and selected time points

Query #9: The authors state that slightly higher HbA1C levels were observed in 60-70 year old population. However, this age group also contained more males than females. Are there known sex differences that may contribute to the higher HbA1C levels?

Response: Authors thank the reviewer for this query.  Please see the data provided in the supporting information regarding the sex differences and variations in HbA1C levels

Query #10: On line 253, the authors state that the HbA1C levels increased to 58.2% from baseline in the case group and that this indicated that the phone calls were successful. However, this conclusion cannot be made unless this increase was compared to the change in HbA1C levels in the control group from the baseline to the end of the study. In addition, all the changes seen in the case group from baseline should be compared to changes seen in the control group to accurately assess the phone-call education system. 

Response: In fact, we have compared the results within the group as well as between the respective groups (Control VS Cases).  For better clarity we have provided these details in the supporting information.

Query #11: In line 272, it states ...data showed a visible difference in the control and diabetic subjects." However, both the control and case groups included diabetic subjects. 

Response: All the participants in this RCT are diabetics.  Differences in the base line knowledge scores in case and control groups were observed in the study participants.  These differences were documented in Table 2.

Query #12: In section 3.3, are the baseline characteristics significantly different between the two groups, as suggested? If so, how do these differences affect the results? P-values here would help clarify these differences. 

Response:  As suggested by the reviewer, the differences mentioned in Table 2 were statistically analyzed and results documented in the revised submission

Reviewer 2 Report

Major concerns:

1. Data is presented in a very difficult way, i.e.:

- figure 1a and 1b are difficult to read and do not add interesting insight into the paper

- a table/graph showing the progression of HbA1c in the case and control group should be added, as it is the main focus of the study

- Table 1 and Table 2 are dense with information, should be simplified and a significance column to highlight differences should be added

- in table 1 the duration of diabetes is stated as mean but reports a range and should be corrected

2. Unit of measure for HbA1c must be added, otherwise it might not be understandable around the world

3. It is not clear the antidiabetic drugs used by the subjects in both case and control group, authors should better describe it as it might have impacted the results

4. In table 1 there are patients in both case and control group which have a recent diagnosis of T2DM (0.08 years): these subjects tend to have an uncompensated disease with a rapid response with therapy. Authors should acknowledge the numerosity of these subjects and exclude them if it affects the overall result

5. In the methods section it is stated that the control group does not attend the education camps, but in results 3.1 line 204 it stated that 5% of controls were unable to attend them: how is it possible?

6. Genderwise analysis paragraph is extremely confusing, but as pregnancy does not appear among the inclusion/exclusion criteria and the big impact it might have on metabolism authors should explain if there are pregnant women in the cohort

7. Drug compliance might have a very big impact on the results, how did the author check for compliance? As per 3.4 line 312-313 a way of assessment seems to be lacking, and this is a major limitation that must be clearly stated in the text

8. Authors should refrain from using approximation in the results section and present just the exact data. A more discorsive lexicon can be used only in the discussion to improve readability

9. There seem to be a big baseline difference between case and control HbA1c, and higher values tend to decrease more rapidly: did the authors check if the two values are statistically significant? That could have greatly affected the results

10. It is not clear how the knowledge on diabetes was assessed

11. Data on the number of subjects should account only for the patients that concluded the trial and not for all the enrolled ones

Minor concerns:

1. Citation 17 in the introduction is confusing (this study refers to the text or ref 17?) and should be moved in the discussion

Author Response

Query #1. Data is presented in a very difficult way, i.e.:

- figure 1a and 1b are difficult to read and do not add interesting insight into the paper

- a table/graph showing the progression of HbA1c in the case and control group should be added, as it is the main focus of the study

- Table 1 and Table 2 are dense with information, should be simplified and a significance column to highlight differences should be added

- in table 1 the duration of diabetes is stated as mean but reports a range and should be corrected

Response:  Figures 1a and 1b are modified as per the reviewer #1 suggestion.  Tables 1 and 2 are also modified to make the data more clearer

Query #2: Unit of measure for HbA1c must be added, otherwise it might not be understandable around the world

Response: HbA1C is measured in the form of percentage. 

Query #3: It is not clear the antidiabetic drugs used by the subjects in both case and control group, authors should better describe it as it might have impacted the results

Response: Authors do agree with the reviewer.  In fact, we have provided the information related to the medication and medication adherence in the Table 2.  No significant differences were observed between control and case groups with reference to medication regimen.

Query #4: In table 1 there are patients in both case and control group which have a recent diagnosis of T2DM (0.08 years): these subjects tend to have an uncompensated disease with a rapid response with therapy. Authors should acknowledge the numerosity of these subjects and exclude them if it affects the overall result

Response:  Authors do agree with the reviewer that there are patients in both case and control group with a recent diagnosis of T2DM.  Since both control (Participants with < 1 year of diabetes n = 12) and case (Participants with < 1 year of diabetes = 10) groups contain similar number of participants, we have not excluded them from the study

Query #5: In the methods section it is stated that the control group does not attend the education camps, but in results 3.1 line 204 it stated that 5% of controls were unable to attend them: how is it possible?

Response: Authors apologize for this mistake.  It is applicable only to cases but not the controls.  The text is updated in the revised submission

Query #6: Genderwise analysis paragraph is extremely confusing, but as pregnancy does not appear among the inclusion/exclusion criteria and the big impact it might have on metabolism authors should explain if there are pregnant women in the cohort

Response:  Authors do agree with the reviewer that pregnancy might influence the diabetes and overall results of this study.  In our study, no pregnant women is present in the cohort.

Query #7: Drug compliance might have a very big impact on the results, how did the author check for compliance? As per 3.4 line 312-313 a way of assessment seems to be lacking, and this is a major limitation that must be clearly stated in the text

Response: Authors do agree with the reviewer that drug compliance will have a great impact on the study outcomes.  As detailed in Table 2, information pertaining to medication regimen / adherence to the medication are listed.  Control and cases have similar medication regimen (P > 0.05).  But, the adherence to prescribed medication is much higher in control group (82.6% responded “Always”) compared to cases (54.07% responded “Always”).  Despite this adherence to medication, the control group did not show much improvement in the HbA1C compared to cases.  The data pertaining to the type of medication and adherence patterns are collected by questionnaire

Query #8: Authors should refrain from using approximation in the results section and present just the exact data. A more discorsive lexicon can be used only in the discussion to improve readability

Response:  As suggested by the reviewer we have avoided using approximation in the results of resubmitted version

Query #9: There seem to be a big baseline difference between case and control HbA1c, and higher values tend to decrease more rapidly: did the authors check if the two values are statistically significant? That could have greatly affected the results

Response:  Authors do agree with this comment.  In fact more number of individuals in the control group are in the range of 5.1-7.0 HbA1C compared to case group in both sexes (19 Vs 3 in case of Males and 13 Vs 3 in case of females, respectively between controls and cases).  Diabetes education through phone calls to cases wherein more number of individuals are having HbA1C >7.1, positively impacted their HbA1C and the knowledge about diabetes management. 

Query #10: It is not clear how the knowledge on diabetes was assessed

Response: Knowledge on diabetes is assessed by using a questionnaire as detailed in methods section

Query #11: Data on the number of subjects should account only for the patients that concluded the trial and not for all the enrolled ones

Response: The final improvement in HbA1C was mentioned by considering the individuals who had completed the study

Minor concerns:

Query #1: Citation 17 in the introduction is confusing (this study refers to the text or ref 17?) and should be moved in the discussion

Response: Authors have updated the text in the revised submission.  Now, the citation 17 is appropriately discussed

Round 2

Reviewer 1 Report

The revisions to the manuscript have clarified and addressed my concerns.

However, the figures are still unclear. In figure 1, the values above each bar graph are unclear in what those values are since the data is presented in the subsequent bar graphs, the unrelated values should be removed. The way the graphs are currently presented suggests that those values are the values of the Y axis, like in figure 2; however, that is not the case. 

Also, in figure 3 and figure 4, the graphs are still unclear on what is being shown, if it is the number of cases per group or the % HbA1C per each group. The most important measurement should be the focus of the graphs. The way it is presented now needs to be clarified what the main readouts are.

Throughout the manuscript, the presentation of the data should be consistent. Sometimes the value above each bar represents the value of the y-axis, and sometimes it is a different unrelated measurement. The way the data is presented should be consistent and graph the most important data.  

Author Response

Reviewer: The revisions to the manuscript have clarified and addressed my concerns.

Authors: Authors would like to thank the reviewer for mentioning that the revised version has clarified and addressed the concerns

Query #1: However, the figures are still unclear. In figure 1, the values above each bar graph are unclear in what those values are since the data is presented in the subsequent bar graphs, the unrelated values should be removed. The way the graphs are currently presented suggests that those values are the values of the Y axis, like in figure 2; however, that is not the case. 

Response:  The values above the bar in Figure 1 are removed in the revised submission

Query #2: Also, in figure 3 and figure 4, the graphs are still unclear on what is being shown, if it is the number of cases per group or the % HbA1C per each group. The most important measurement should be the focus of the graphs. The way it is presented now needs to be clarified what the main readouts are.

Response: The data presented in figures 3 and 4 are re-plotted to make the figures more clearer. 

Query #3: Throughout the manuscript, the presentation of the data should be consistent. Sometimes the value above each bar represents the value of the y-axis, and sometimes it is a different unrelated measurement. The way the data is presented should be consistent and graph the most important data.  

Response: Authors would like to thank the reviewer.   In the revised submission, the data is presented in a consistent form throughout the manuscript
